# The effectiveness and efficiency of using normative messages to reduce waste: A real world experiment

Gabby Salazar[1☉], João Neves[2☉]*, Vasco Alves[2], Bruno Silva[2], Jean-Christophe Giger[3], Diogo Veríssimo[4]

1 School of Forest, Fisheries, and Geomatics Sciences, University of Florida, Gainesville, Florida, United States of America, 2 Department of Science and Education, Zoomarine, Algarve, Albufeira, Portugal, 3 Psychology Research Centre (CIP), University of Algarve, Faro, Portugal, 4 Department of Zoology, University of Oxford, Oxford, England, United Kingdom

☉ These authors contributed equally to this work.
* joao.neves@zoomarine.pt

**Data Availability Statement:** We have made the data available. You can now find the dataset here: https://figshare.com/s/648611d23cb654d0dfcf.

## Abstract

Although they are only home to 16% of the global human population, high-income countries produce approximately one third of the world's waste, the majority of which goes to landfills. To reduce pressure on landfills and natural systems, environmental messaging should focus on reducing consumption. Messages that signal social norms have the potential to influence people to reduce their consumption of comfort goods, such as straws, which are not a necessity for most people. We conducted a randomized field-experiment at a marine park in Portugal to test whether different normative messages reduced visitors' paper straw use when compared to non-normative messages. We found that a message framed around a positive injunctive norm significantly reduced straw use compared to a non-normative message. We estimated that using the message at 17 park concession stands could keep over 27500 straws out of landfills annually and save the park money after two years.

## Introduction

Humans have a global waste problem. High-income countries, which are only home to 16% of the world's human population, generate about 34% of the world's waste each year [1]. In 2018, European Union countries generated 2337 million tons of waste across all economic activities and households [2]. That is approximately five tons of waste per resident of the European Union. Concerningly, only 54.6% of waste was treated in recovery operations, including 37.9% that was recycled, while the remaining 45.4% was either sent to landfills, incinerated, or disposed of otherwise [2]. In environmental circles, the three R's: reduce, reuse, recycle have long been a common mantra, with most emphasis placed on recycling [3]. However, recycling alone will not solve our global waste problem [4]. To reduce pressure on natural systems and on landfills, messaging should focus on reducing consumption and keeping waste out of the global trash cycle. This can be accomplished by encouraging people to reduce, reuse and refuse, rather than consume and recycle [5].

The public DOI is: https://doi.org/10.6084/m9.figshare.15134754.v1.

**Funding:** This work was funded by national funds through Fundação para a Ciência e a Tecnologia (FCT) as part the project CIP - Refª UID/PSI/04345/2020 (Jean-Christophe Giger). The funders had no role in study design, data collection and analysis, decision to publish, or preparation of the manuscript.

**Competing interests:** The authors have declared that no competing interests exist.

Single-use plastics have recently become a hot button issue because of their contribution to waste and plastic pollution [6]. In 2015, a video of a distressed sea turtle with a straw being removed from its nostril went viral and became an emblem of the anti-straw movement [7]. Plastic straws, which are difficult to recycle, have been vilified because they often end up in landfills or in the ocean and other waterbodies [8]. A recent study estimated that only 9% of global plastics that have been produced have been recycled as of 2015 [9]. This has led some governmental bodies, including the European Union, to ban single-use plastics, including plastic straws [10]. Instead of rethinking the need for straws, many companies are now switching to paper and cardboard alternatives [11].

While the switch from plastic to paper straws is more sustainable, exchanging one disposable good for another will not solve all environmental problems [11]. For the majority of users, straws are a type of comfort good, a good that provides some benefit, but is not a necessity [12]. While some groups of disabled people need straws [13], the majority of people could forego the use of straws, reducing their contribution to the world's waste problem. There is an urgent need to understand how to persuade people to reduce their use of unnecessary comfort goods, such as disposable straws.

Environmental groups are increasingly using insights from behavioral science to nudge people toward more pro-environmental behaviors, including waste reduction [14]. A meta-analysis that reviewed behavior-change interventions related to waste production found evidence supporting the use of defaults and commitments to lower amounts of paper, plastic, and food waste [15]. The use of social norm messaging, however, showed mixed results. While social norms reduced plastic bag use and food waste, only one of the three interventions that used norms to reduce paper waste had a significant effect [15–18]. Further research is needed to understand which types of behavioral interventions can effectively reduce the use of paper products.

Normative messages, or messages that signal social norms, have been widely used to influence human behavior [19, 20]. Social norms are unwritten rules about acceptable behaviors in particular settings [21]. These norms have powerful effects on human behavior because people are driven to conform to local customs [21]. In a field experiment, hotel guests were significantly more likely to reuse towels when they received a message stating that 75% of other hotel guests reuse their towels, than when they received a more generic message that did not signal any norms [22]. Normative messaging has been widely used to promote pro-environmental behaviors, such as reducing residential energy and water use [23, 24] and choosing sustainable transportation [25, 26].

Different types of social norms can be activated through messages. Descriptive social norms describe our perceptions of how other people typically behave [27, 28]. While descriptive norms can promote pro-environmental behaviors, they can also reinforce unsustainable behaviors. Cialdini et al. (2006) tested different normative messages as part of campaign to stop visitors from stealing petrified wood from a protected area. They found that messages that described how much wood other visitors were taking tended to increase theft.

Injunctive social norms refer to our perceptions of how others think we should behave [27, 28]. Messages framed around injunctive norms typically express approval or disapproval of certain behaviors. Cialdini et al. (2006) [29] found that a strong injunctive message ("Please don't remove petrified wood from the park"), resulted in less theft of petrified wood from a protected area than other message conditions. Similarly, de Groot et al. (2013) [30] found that an injunctive message in a supermarket resulted in significantly lower plastic bag use among shoppers than a non-normative message about the environment.

Moral norms are the rules of morality that people in a certain society or group are expected to follow [31]. Some groups view the protection of the planet as a personal responsibility and a

moral obligation [32]. For example, some private landowners in the U.S. feel a moral obligation to prevent the extinction of endangered animals [33] and many households in Sweden feel a moral obligation to recycle [34]. Moral norms have received less attention than descriptive and injunctive norms in tests of environmental messaging and may offer a powerful force for promoting environmental actions [35].

Studies examining the relative influence of different types of normative messages have shown that their influence varies across contexts [36, 37]. Whether a message is negatively worded or positively worded is also likely to influence perceptions and behaviors [29, 38]. For example, a positively worded injunctive message that expresses approval of a behavior might have a different effect than a negatively worded injunctive message that expresses disapproval of the opposite behavior.

While many field-experiments have tested the influence of normative messages on water conservation, energy conservation, and littering, few have examined the extent to which normative messages can influence people to reduce consumption of comfort goods, such as a straws [15, 37, 39]. Additionally, a recent meta-analysis of 91 field-experiments that used social norms to promote pro-environmental behaviors found that only 24 experiments took place in a European country other than the United Kingdom [37]. It is important to study messaging in different countries and contexts because culture may influence how people respond to message frames [40]. We conducted two randomized field-experiments at a marine park in Portugal to test whether different normative messages reduced visitors' paper straw use when compared to non-normative messages. We hypothesize that normative messages will result in significantly less straws taken than non-normative messages.

## Study 1: Pilot experiment

**Materials and methods.** The two experiments took place at Zoomarine, a marine park located in Algarve, the southernmost region of continental Portugal (https://www.zoomarine. pt/en/) (Fig 1). Zoomarine provides entertainment and environmental education to visitors through a combination of marine-inspired educational exhibits and amusement rides. The pilot study (Study 1) sought to identify which type of social norm messaging (injunctive, descriptive, or moral) most effectively nudged park visitors to reduce their use of paper straws when purchasing beverages from a park concession stand. The full experiment (Study 2) built on the results of Study 1 by testing the most effective social norm messages from Study 1 against a control message. This study was approved by the University of Florida's Internal Review Board (IRB202002244) and the need for consent was waived. IBM SPSS Statistics 26 was used to calculate descriptive statistics for both experiments to compare the mean ratios of different conditions.

One concession stand in the park was selected for this pilot study and paper straw dispensers and message signs were installed close to the concession stand cashier. We piloted six different message conditions and a control condition over 72 days between July 1st and September 10th, 2018 (Table 1). Each message condition was in place for three days at a time and a control condition (no message) was displayed for three days in between each message condition. All social norm message conditions were on display for two three-day periods over the course of the pilot. Messages were displayed in both English and Portuguese.

Data on the number of drinks sold to visitors and the number of paper straws taken by visitors were recorded each day and were used to calculate the ratio of paper straws taken to drinks sold per day.

**Results.** The experiment ran for 72 days; operator error occurred on 7 of these days, resulting in 65 days of accurate data collection. Operator error means that there were mistakes

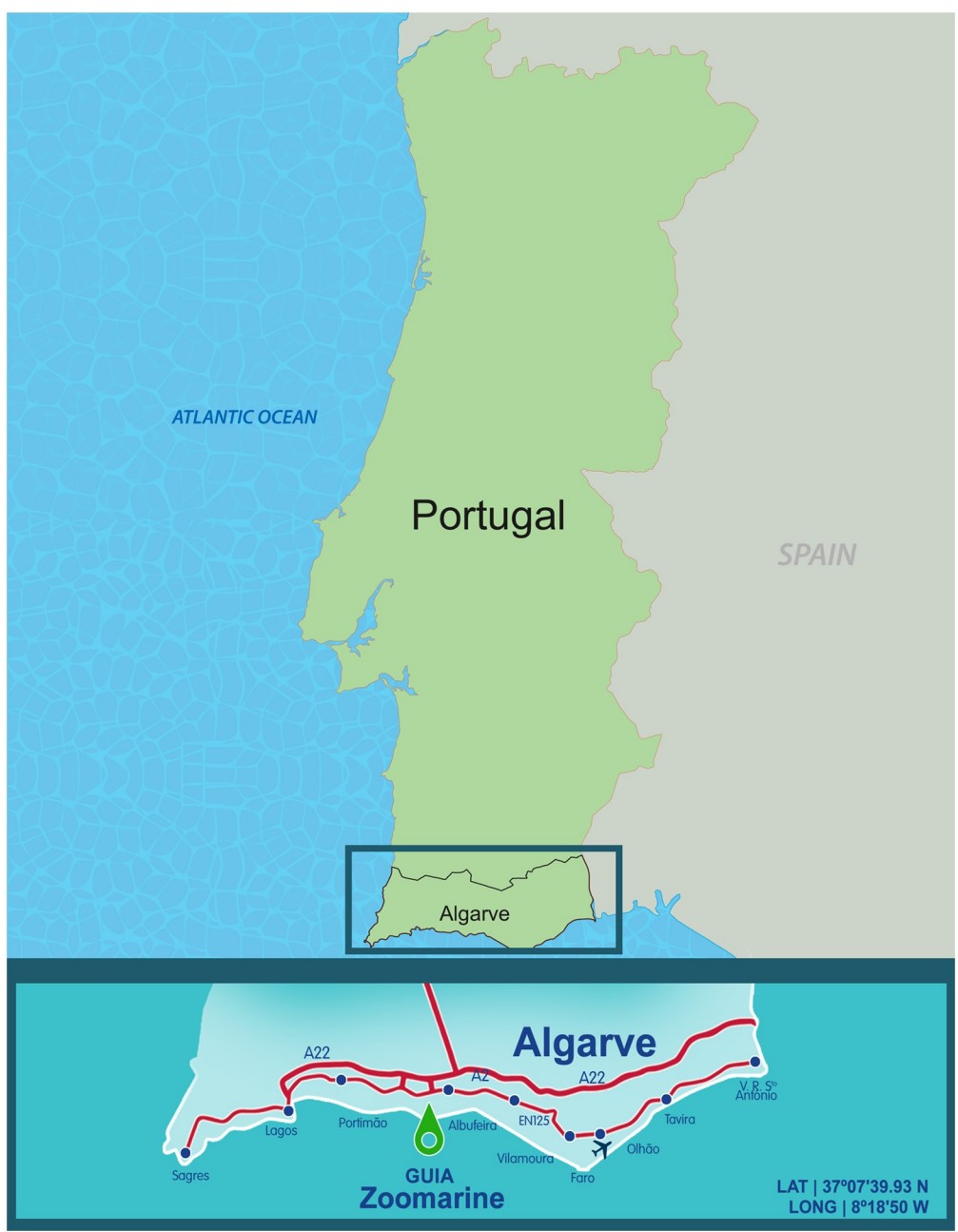

**Fig 1. Map of Portugal, which shows where Zoomarine is located.** Reprinted from Zoomarine under a CC BY license, with permission from Zoomarine.

in the data collection on a particular day that rendered the data unusable. Errors included the implementation of incorrect message signs on a particular day, the temporary absence of straws in a dispenser due to shortages, and errors due to cashier shift turnover. In total, 15,279 drinks were sold over the 65 days and 4,684 paper straws were taken. The ratio of straws taken to drinks sold was calculated for each day and the mean ratio for each condition was calculated (Table 2).

**Table 1. Messages tested during the pilot experiment.**

| Condition | Message displayed | Number of days displayed |
|---|---|---|
| C—Control | No information | 36 |
| (N1)—Negative descriptive social norm | 80% of our visitors choose not to use disposable straws with their drinks. The planet thanks you! | 6 |
| (N2)—Positive descriptive social norm | 80% of our visitors choose to drink directly from the cup or can. The planet thanks you! | 6 |
| (N3)—Negative injunctive social norm | Choose not to use disposable straws with your drink. The planet thanks you! | 6 |
| (N4)—Positive injunctive social norm | Choose to drink directly from the cup or can. The planet thanks you! | 6 |
| (N5)—Negative moral social norm | Protecting our planet is our duty. Choose not to use disposable straws with your drink. | 6 |
| (N6)—Positive moral social norm | Protecting our planet is our duty. Choose to drink directly from the cup or can. | 6 |

The data did not meet the assumptions for a one-way ANOVA. Two conditions had outliers, as assessed by inspection of a boxplot: the negative injunctive norm condition had a single outlier, while the control condition had three outliers, two of which were extreme outliers (more than 3 box-lengths away from the edge of the box). The results of a Shapiro-Wilk's test demonstrated that the data were not normally distributed for two conditions: control (p = .001) and negative injunctive norm (p = .034). There was homogeneity of variances, as assessed by Levene's test for equality of variances ($p$ = .493). Because the data violated these assumptions, a Kruskal-Wallis test was run to determine if there were differences in ratios between the seven conditions. The mean rank of ratios was not statistically significantly different between groups, $\chi^2(6)$ = 7.971, $p$ = 0.240, $\varepsilon^2$ = .125 [41]. This is a moderate effect size. Recognizing that a one-way ANOVA is somewhat robust to violations of these assumptions [42], we also ran a one-way ANOVA and found that there was still no difference in ratios between the seven conditions, $F(6,58)$ = 1.315, $p$ = .265, est. $\omega^2$ = .029. This is a relatively small effect size.

## Study 2: Full experiment

**Materials and methods.** Building on the results of the pilot, a full experiment was conducted in the summer of 2019. The same concession stand was used for the full experiment, but straws were no longer available in the straw dispenser for free withdrawal due to a larger strategy across the park to nudge visitors to reduce consumption. Visitors now had to ask the

**Table 2. Mean ratio of straws taken to drinks sold for six message conditions and a control.**

| Condition | N° days accurate data collected | Ratio of straws taken to drinks sold x̄ (SD) |
|---|---|---|
| C—Control | 30 | 0.347 (0.155) |
| (N1)—Negative descriptive social norm | 6 | 0.318 (0.129) |
| (N2)—Positive descriptive social norm | 6 | 0.250 (0.147) |
| (N3)—Negative injunctive social norm | 6 | 0.215 (0.041) |
| (N4)—Positive injunctive social norm | 6 | 0.262 (0.097) |
| (N5)—Negative moral social norm | 6 | 0.272 (0.143) |
| (N6)—Positive moral social norm | 5 | 0.249 (0.131) |

cashier for a straw. The message displays were located in a clearly visible area on the counter near the cashier.

The experiment ran for a total of 87 days between June 24th, 2019 and September 20th, 2019 to determine whether different messages had an influence on the ratio of paper straws taken to drinks sold at a concession stand in the park. A message pair using positive and negative injunctive norms was selected for broader testing based on the results of the pilot, which showed that the mean ratio of straws taken to drinks sold was lowest for the negative injunctive social norm condition ($\bar{x} = 0.215$; $SD = 0.041$). Given the lack of a significant result in the pilot study, the choice to move forward with the injunctive message pair was mainly informed by studies that suggest that injunctive norms are more effective than descriptive norms at promoting desired environmental behaviors [28, 29].

Three messaging conditions were tested during the experiment, each in place for 3 days at a time:

N3 Condition: *Negative injunctive social norm*
  Choose not to use straws with your drink.

N4 Condition: *Positive injunctive social norm*
  Choose to drink directly from the glass or can.

Condition: *Control*
  We are changing! 85% of all our disposable materials are already environmentally sustainable.

Messaging conditions were randomly assigned to different three-day periods over the course of the experiment. Data on the number of drinks sold to visitors and the number of paper straws taken by visitors were recorded each day and were used to calculate the ratio of straws taken to drinks sold per day.

*Calculation of waste avoided and intervention cost.* We estimated the amount of waste that could be avoided through the intervention over the course of a year by comparing the average number of straws taken during the control condition to the average number of straws taken during the positive injunctive norm condition and extrapolating these figures over the 250 days that the park is open on average each year and over the 17 concession stands in the park. We also calculated the cost of running the intervention and experiment, including the fixed cost of staff time to design and promote the intervention, the fixed cost of the displays, and the variable cost of researcher time counting straws, entering data and changing displays each day. We then calculated the cost-benefit ratio for the intervention over time to determine when the park would see a return on investment. We also calculated the cost to avoid one kilogram of waste over time by dividing the net cost of the intervention each year by the number of kilograms of straws avoided each year.

**Results.** The experiment ran for 87 days; operator error occurred on 10 of these days, resulting in 77 days of accurate data collection. Operator error means that there were mistakes in the data collection on a particular day that rendered the data unusable. In total, 11,346 drinks were sold over the 77 days and 1,597 paper straws were taken. Over the course of the experiment, data were collected for positive injunctive norm message for 25 days, for the negative injunctive norm message for 26 days, and for the control message for 26 days.

A Welch's ANOVA was conducted to determine if the mean ratio of paper straws taken to drinks sold was different for the different messaging conditions because the data did not meet all the assumptions for a one-way ANOVA. There were no outliers in the data, as assessed by inspection of a boxplot for values greater than 1.5 box-lengths from the edge of the box. The results of a Shapiro-Wilk's test also demonstrated that the ratio values were normally

distributed for the negative injunctive norm condition ($p = 0.101$) and the control condition, ($p = 0.077$), while the distribution of ratio values for the positive injunctive norm condition was on the edge of normal distribution ($p = 0.05$). The assumption of homogeneity of variances was violated, as assessed by Levene's test for equality of variances ($p = 0.004$), which led to the use of a Welch's ANOVA. The mean ratios of paper straws taken to drinks sold were statistically significantly different between the different conditions, *Welch's F*(2, 45.39) = 5.85, $p = 0.006$, est. $\omega^2 = 0.112$. This is a relatively large effect size.

The mean ratio of straws taken to drinks sold per day decreased from the control condition ($\bar{x} = 0.172$, *SD* = 0.094), to the negative injunctive norm condition ($\bar{x} = 0.152$, *SD* = 0.076), to the positive injunctive norm condition ($\bar{x} = 0.111$, *SD* = 0.045), in that order (Table 3). A Games-Howell post hoc analysis revealed that the mean decrease from the positive injunctive norm condition to the control condition (-0.062, 95% CI [-0.1116, -0.0114]) was statistically significant ($p = 0.013$), *Cohen's d* = 0.835 (Fig 2). In this case, we reject the null hypothesis and accept the alternative hypothesis that a positive normative message resulted in significantly less straw use than a non-normative message. The mean decrease from negative injunctive norm condition to the control condition (-0.0208, 95% CI [-0.0782, 0.0366]) was not significant ($p = 0.659$), *Cohen's d* = 0.243. The mean difference between the negative injunctive norm and the positive injunctive norm was not significant (0.0407, 95% CI [-0.0017, 0.0832], $p = 0.063$), *Cohen's d* = 0.649.

**Table 3. Mean ratio of straws taken to drinks sold for two message conditions and a control.**

| Condition | N˚ days data collected | N˚ straws taken | N˚ drinks sold | Ratio of straws taken to drinks sold x̄ (SD) |
|---|---|---|---|---|
| Positive injunctive social norm | 25 | 3888 | 431 | 0.111 (0.045) |
| Negative injunctive social norm | 26 | 3611 | 548 | 0.152 (0.076) |
| Control | 26 | 3847 | 618 | 0.172 (0.094) |

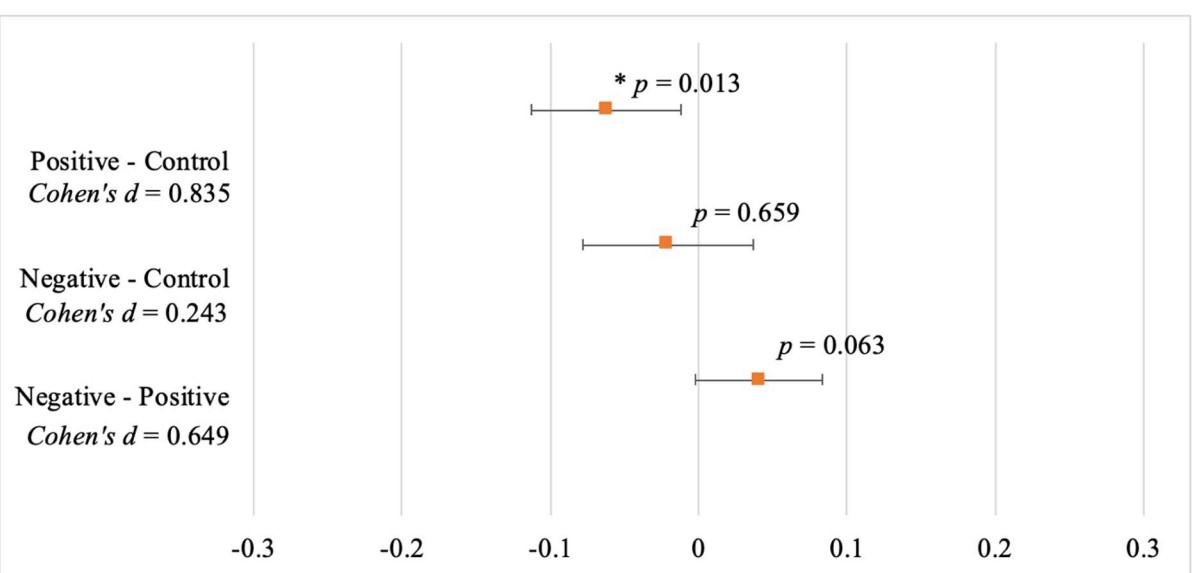

**Fig 2. Results of a Games-Howell post-hoc analysis shows the mean differences between conditions with 95% confidence intervals.** *Note.* * Denotes a statically significant difference at the 0.05 level between groups.

*Calculation of waste avoided through intervention.* Over the 26 days that the control message was in place, an average of 23.77 straws were taken per day. Over the 25 days of the positive injunctive norm message was in place, an average of 17.24 straws were taken per day. We estimated the number of straws saved per year at one concession stand by multiplying the average number of straws taken per day for each condition by the 250 days that the park is open on average annually: 5942.5 straws would be taken if the control message was used and 4310 straws would be taken if the positive injunctive norm message was used. If the positive injunctive norm message was used across the 17 concession stands in the park, the park could potentially keep 27752.5 straws out of its waste stream each year. Paper straws weigh approximately 1.1 grams each [43], which means that 30.53 kilograms of trash removed from the waste stream each year.

The experiment had two fixed costs amounting to 685€. This figure includes researcher's time to design and promote the project (640€) and the cost to produce and design signage (45€). The variable cost of counting straws, entering data, and changing displays amounted to 1.5€ per day at the concession stand. In total, it cost approximately 815.5€ to run the full experiment for 87 days at a single concession stand. To expand the intervention to the other 16 stands in the park, there would be a fixed cost of 480€ (including 30€ per stand for display signage). The total cost in Year 1 would be 1295.50€. Straws cost 0.019€ each, which means that 527.30€ could also be saved on straws annually.

$$\text{Cost-benefit ratio for Year 1 } (1295.5/527.3) = 2.46$$

$$\text{Cost-benefit ratio for Year 2 } (1295.5/1054.6) = 1.23$$

$$\text{Cost-benefit ratio for Year 3 } (1295.5/1581.9) = 0.82$$

In Year 3, the park would save 286.40€ on the purchase of straws, with a return on investment of 22.11%. The cost of avoiding one kilogram of waste would be 25.16€ in Year 1 and 7.89€ in Year 2. In Year 3, the financial benefits of the intervention would outweigh the costs and the park would save 9.38€ for every kilogram of waste avoided.

## Discussion

This field experiment measured actual behavior as well as randomly assigned experimental and control conditions in a public setting. We tested whether normative messages could reduce visitors' use of paper straws at a marine park and found that the positively worded injunctive message resulted in significantly less paper straw use than the non-normative message and that there was a relatively large effect size. We also found that the intervention could be a cost-effective way to reduce waste and could potentially save the marine park money after only two years.

It is surprising that a positive injunctive norm message was most effective. Past research on environmental messaging suggests that negatively worded injunctive messages may have a greater impact on behavior than positively worded injunctive messages [29]. Similarly, a recent study on the promotion of healthy food consumption found that a negative injunctive message was more effective than a positive injunctive message [38]. Negative information is often afforded greater attention and weight in a person's consciousness than positive information [44, 45]. Our study, which found the positively framed message to be more effective, suggests that further research is needed on how framing influences the effects of injunctive norms [38]. It is important to acknowledge that we only tried one version of a message for each norm condition and that our sample sizes were limited, which means the randomization process may not have fully balanced all variables (e.g., respondent profile) across all conditions. It is possible that different messages within the different conditions could have influenced behavior in other

ways. Framing effects are also likely to vary across cultures and countries. When examining framing effects across seven Central and Western European countries, one study found differences in the way that consumers from different countries, even within the same region, responded to positive and negative message frames [40].

While we did not collect information on participant characteristics, which could help better explain our results, the psychological literature provides some potential explanations as to why the positive injunctive message may have been more effective. One possible explanation is that the negatively worded message may have triggered psychological reactance in some visitors [46]. Reactance is a form of motivational arousal that occurs when someone feels that their personal freedoms are threatened by rules and restrictions [46]. Resistance can occur when injunctive social norms (or proscriptive messages) appear to impinge on personal freedoms and may even lead people to increase the undesirable behavior [47]. While this is speculative, reactance may have decreased the effect of the negative injunctive message, effectively erasing any signal when compared to the control group.

The positive injunctive message may have been more effective because it reminded visitors of their pro-environmental attitudes and values. While some visitors to zoos and aquariums come solely for entertainment, many visitors are interested in learning about animals [48] and about ways to make a difference for conservation [49]. The positive injunctive message may have reminded visitors of their pre-existing environmental attitudes, nudging them to act in a way that is consistent with these attitudes. Self-perception theory suggests that people are generally motivated to maintain consistency in their behaviors and beliefs [50]. Cornelissen et al. (2008) [51] found that cueing behaviors as pro-environmental increased the likelihood that people would engage in those environmental behaviors, particularly if people perceived of themselves as being environmentally responsible. It would be interesting to test similar messages at a concession stand in a setting that has no environmental associations, such as a sporting event or a musical concert.

When comparing the positive and injunctive norm messages, salience bias might also help explain why the negative injunctive message resulted in higher paper straw use than the positive injunctive message [52]. The negative injunctive message included the word 'straw' while the positive injunctive message did not. Simply seeing the word could have triggered some people to think of straws and to ask for one [52]. Framing theory suggests that even minor changes in the presentation of an issue can result in significant changes in how that issue is perceived [53]. In future experiments, matching the positive injunctive and negative injunctive messages as closely as possible could help eliminate any influence of word choice and sentence structure on outcomes.

The experimental setting may also have influenced the results. The Focus Theory of Normative Conduct suggests that normative elements are likely to more be effective if they are focal, or salient, when the behavioral decision is being made [27]. In this experiment, explicit normative messages were prominently placed in the behavioral setting, right next to the cashier. The broader saliency of the issue in people's minds could have also influenced behavioral decisions. Straws have been in the Portuguese news and the international news over the past few years as pressure has mounted for people around the world to reduce their use of single-use plastics [54]. Given the widespread discussion around straws, it is possible that visitors had already formed associations with straws (e.g., using straws is not environmentally friendly) that influenced their behavior.

In the full experiment, the fact that people were required to ask the cashier for a straw, rather than passively taking straws from a dispenser, may have also influenced results. This change also means it is not possible to compare effect sizes between the pilot experiment and the full experiment, which is a limitation of the study. Changes in default options can have a

strong influence on behavior [55, 56]. Having to request a straw could also amplify social norm effects because perceived social disapproval can generate strong effects on behavior [57]. A field experiment found that people were more likely to choose reusable takeaway boxes when they witnessed others using a reusable takeaway box [58]. In this experiment, asking for a straw after seeing the signage required visitors to directly flout explicit social norms, both in front of the cashier and in front of other customers in line. Having to ask for a straw may also disrupt automated choices, increasing the likelihood that consumers take the time to think about whether they actually need a straw [59].

Future research at Zoomarine could also explore spatial and temporal variations by carrying out this experiment at more stands for a longer period of time. Studies could also explore how social norms can influence the consumption of other comfort goods, including other single-use products. It would also be interesting to examine the interactions between positive and negative framing and social norms in more detail. Beyond Zoomarine, more real-world experiments should test how environmental messaging can influence people's environmental behaviors [60] and how normative messages are affected by framing [38].

*Costs and benefits of interventions*. Moving forward, it is critical that more field behavioral experiments examine the financial efficiency of interventions, as well as their effectiveness [61]. Interventions are more likely to be adopted by companies and governments if they make financial sense. In their review of interventions designed to increase pro-environmental behaviors, Byerly et al. (2018) found that only 15 of the 72 studies they reviewed examined cost-effectiveness. Similarly, few studies have examined the potential cost savings of social norm interventions [61, 62].

We estimated that displaying the positive injunctive message at 17 park concessions stands could keep approximately 27500 straws out of the park's waste stream each year. After an initial investment of 815.5€ to run the experiment, the only cost to implement this intervention would be the cost of the displays at each concession stand. Our cost-benefit calculation is based on some key assumptions, including the assumption that all concession stands will sell similar numbers of drinks and the assumption that visitation rates and visitor behavior will remain similar. While some events, such as the COVID-19 pandemic, show that this is not always the case, we feel that these estimates are still informative. Accepting these assumptions, our calculations project that the return on investment could be approximately 22% if the intervention ran for three years, and that rate would continue to increase over time. The cost to avoid one kilogram of waste would also drop over time, from approximately 25€ in Year 1 to 8 € in Year 2. The few waste-related interventions that include social norms and cost-benefit calculations indicate that these campaigns are a promising direction for waste reduction. In Port Colborne, Ontario, Canada, a campaign and program to divert organic waste from a landfill had a fixed cost of $269,500 and an ongoing annual cost of $23,000. Even with these costs, the campaign had a four-year pay-back period and a return on investment of approximately 12% over the first ten years [63]. Similarly, a program to reduce household energy consumption found that norms-based messaging could reduce electricity consumption in the average household by over 2% in a randomized control trial [61]. The study calculated the cost effectiveness of the program and showed that it compared favorably to the estimated cost effectiveness of similar energy-efficiency programs. More businesses might be convinced to implement interventions to reduce waste if they understood the potential costs and benefits over time.

## Conclusions

Across the globe, there is an urgent need to find strategies to reduce waste production. Our research demonstrates that minor changes in the wording of a normative message can

significantly influence behavioral outcomes, moving individuals and companies toward more sustainable practices. While straws may seem like a minor contribution to the waste stream, a report estimated that the countries of the European Union consumed 36.4 billion drinking straws annually [64]. Furthermore, interventions like ours could potentially reduce waste from other comfort goods, such as takeout containers and single-use bags, in ways that not only support environmental sustainability but also make financial sense.

## Acknowledgments

We would like to thank Zoomarine for allowing us to run this experiment at the marine park.

## Author Contributions

**Conceptualization:** João Neves, Diogo Veríssimo.

**Data curation:** João Neves.

**Formal analysis:** Gabby Salazar.

**Funding acquisition:** Jean-Christophe Giger.

**Investigation:** João Neves, Vasco Alves, Bruno Silva.

**Project administration:** João Neves.

**Supervision:** Diogo Veríssimo.

**Visualization:** Gabby Salazar.

**Writing – original draft:** Gabby Salazar.

**Writing – review & editing:** João Neves, Jean-Christophe Giger, Diogo Veríssimo.

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
