## [Decision Letter · Decision Letter 0]

15 Oct 2021

PONE-D-21-25759The effectiveness and efficiency of using normative messages to reduce waste: A real world experimentPLOS ONE

Dear Dr. Neves,

Thank you for submitting your manuscript to PLOS ONE. After careful consideration, we feel that it has merit but does not fully meet PLOS ONE’s publication criteria as it currently stands. Therefore, we invite you to submit a revised version of the manuscript that addresses the points raised during the review process.

We look forward to receiving your revised manuscript.

Kind regards,

Valerio Capraro

Academic Editor

PLOS ONE

Additional Editor Comments (if provided):

I have now collected two reviews from two experts in the field, whom I thank for their detailed and thoughtful reviews. Both reviewers found the paper interesting and easy to read. However they both expressed a number of concerns that should be addressed in a major revision. Therefore, I would like to invite you to revise your paper following their comments. Needless to say that all comments should be addressed or rebutted. When uploading your revised manuscript, please upload also a response letter containing a point-by-point response to all the reviewers' comments.

I am looking forward for the revision.

Journal Requirements:

Could you therefore please include the title page into the beginning of your manuscript file itself, listing all authors and affiliations

3. Please clarify in your Methods section that IRB waived the need for consent

“This work was funded by national funds through Fundação para a Ciência e a Tecnologia (FCT) as part the project CIP - Refª UID/PSI/04345/2020 (Jean-Christophe Giger)”

6. We note that Figure 1 in your submission contain map images which may be copyrighted. All PLOS content is published under the Creative Commons Attribution License (CC BY 4.0), which means that the manuscript, images, and Supporting Information files will be freely available online, and any third party is permitted to access, download, copy, distribute, and use these materials in any way, even commercially, with proper attribution. For these reasons, we cannot publish previously copyrighted maps or satellite images created using proprietary data, such as Google software (Google Maps, Street View, and Earth). For more information, see our copyright guidelines: http://journals.plos.org/plosone/s/licenses-and-copyright.

    1. You may seek permission from the original copyright holder of Figure(s) [#] to publish the content specifically under the CC BY 4.0 license. 

Reviewers' comments:

Reviewer's Responses to Questions

**Comments to the Author**

1. Is the manuscript technically sound, and do the data support the conclusions?

Reviewer #1: Yes

Reviewer #2: No

2. Has the statistical analysis been performed appropriately and rigorously? 

Reviewer #1: Yes

Reviewer #2: No

3. Have the authors made all data underlying the findings in their manuscript fully available?

Reviewer #1: No

Reviewer #2: No

4. Is the manuscript presented in an intelligible fashion and written in standard English?

Reviewer #1: Yes

Reviewer #2: Yes

5. Review Comments to the Author

Reviewer #1: The authors conducted two social norm intervention studies aimed at reducing consumption of paper straws at one concession stand in a marine park. The first study was a shorter pilot, with the goal of narrowing down the type of social norm messages that would be included in the larger second study. The second study compared a control message, a positive injunctive norm message, and a negative injunctive norm message. Results indicated that the positive injunctive norm message significantly reduced paper straw consumption compared to the control condition. For each day of data collection, the authors collected a count of the number of drinks that were sold at the concession stand as well as a count of the number of paper straws that were requested. These values were transformed into a ratio and each condition was associated with a single value: the mean of these ratios across all days of data collection in that condition. Finally, the authors provided a cost-benefit analysis that extrapolated the results of this study to determine how much money this social norm intervention would yield the marine park over time.

I thought this manuscript was clearly presented and easy to read. The introduction has a nice overview of relevant literature and the discussion section thoughtfully brings up many other psychological factors that could be contributing to the results.

Major Comments:

The use of ratios obscures differences in base rates that may have varied by condition. For example, were there any differences in the number of drinks that were sold across conditions? Relatedly, was any other information about the day recorded? Number of visitors? Temperature? I’m wondering if it’s possible to rule out environmental factors that may have covaried with condition.

I would be interested in seeing what the data that will be made available upon publication looks like. Will it simply be the averages of the ratios for each condition? Or will the data file include the number of drinks ordered and the number of straws requested for each day in each condition? I think the latter would be more useful for people interesting in further exploring this interesting dataset.

Minor Comments:

Page 2, line 30: “Concerningly, only 37.8% was recycled, while 45.7% was sent to landfills.” What accounts for the remaining 16.5%?

Typo page 2 line 39: “Single-use plastics have recently become a become a hot button issue…” ‘become’ is repeated twice.

Page 8: “A message pair using positive and negative injunctive norms was selected for broader testing based on the results of the pilot, which showed that the mean ratio of straws taken to drinks sold was lowest for the negative injunctive social norm condition (x̄ = 0.214; SD = 0.041).” However, the results of the pilot study indicate that the negative injunctive social norm condition was not statistically different from any other condition (including the control condition). Might be more clear that the choice of the injunctive norms was based on the literature because the pilot study didn’t provide any clear winners.

Describe the nature of the operator error for both Study 1 and Study 2.

Why do you use different statistical tests to answer the same question for Study 1 and Study 2? Study 1 uses a Kruskal-Wallis test to determine if there was a difference in ratios by condition. Study 2 uses a Welch’s F test to determine if there was a difference in ratios by condition. Either use the same test for each or provide a justification for why one is more appropriate for each case.

Page 11: When you inspect the normality of the distributions for each condition, are you looking at the distribution of ratio values over the 25-26 days for each condition? Regardless, make it clear what the values are that you are inspecting.

Page 12 “In this case, we reject the null hypothesis and accept the alternative hypothesis that normative messages resulted in significantly less straw use than non-normative messages.” Should be more specific and say ‘a positive normative message’.

Looks like the biggest intervention happened when the park decided that patrons had to ask for straws (a change that happened between Study 1 and Study 2). Would be good to discuss the effect sizes of the norm intervention in terms of the change in control condition values from Study 1 to Study 2.

In Discussion section, be clear that you only tried ONE version of a message for each norm condition. It’s possible that different messages that fit within the different norm definitions may have impacted behavior differently.

The reactance suggestion in the discussion section would predict that negatively worded normative messages would encourage MORE straw consumption than the control condition. This did not happen.

Page 16: “salience bias might also help explain why the negative injunctive message resulted in higher paper straw use” Be clear that you mean higher than the positive injunctive message, not higher than the control condition.

Reviewer #2: This paper described a simple experiment testing the effectiveness of normative messaging on the use of straws. Overall, it was clear and easy to read, however I felt the paper was missing many details to help us understand what was done, why, and the relevance of the findings. I outline these below.

Main concerns

- Why is the data not provided to the reviewers? Data availability is required to publish in PLOS one and is highly relevant for the review process. Please make this available.

- I would like to see the results of the pilot study below the study description. Understanding what was found in that experiment is important to understand the methods of the main experiment of the paper.

- The authors did not find any difference between conditions in the pilot study

- Why use a Kruskal-Wallace test for the pilot? The non-parametric test was justified for the main experiment, but not the pilot. If it passes parametric assumptions, a two-way ANOVA would be appropriate (except for control condition), where you compare Positive/Negative manipulation, and manipulation type (descriptive, injunctive, social norm). Additionally, why not compare each individual group to the control, as you did in the main experiment? This result section was lacking, in both justifications and statistics.

- All test decisions should be justified through the manuscript

- What is operator error? Describe it in detail (can be in a supplement)

- Effect sizes should be interpreted throughout the manuscript

- Data was collected over time, why was this not analysed? Was the straw to drink ratio consistent over time, or did it increase/decrease? This analysis is necessary to make any argument about effectiveness of the intervention. Without it, the discussion about savings over time is premature.

- Why are there no comparisons to other stands in the park over the same time period?

- The authors state this experiment is important because it is conducted in a European country and has been understudied, yet does not discuss cultural effects on messaging. This should be discussed in the introduction and in the discussion – it is very possible that the effectiveness of the messages (positive/negative) may be related to cultural factors

Minor comments

- Describe the Welch’s F test as non-parametric (line 243)

- Interpret effect size of Welch’s F test (line 245)

- Give actual numbers in Table 3, in addition to what is presented (number of drinks sold, number of straws taken)

- Acknowledge the discussion section on reactance as speculative

6. PLOS authors have the option to publish the peer review history of their article (what does this mean?). If published, this will include your full peer review and any attached files.

Reviewer #1: No

Reviewer #2: No

---

## [Author Response · Author response to Decision Letter 0]

18 Nov 2021

Reviewer #1: 

The authors conducted two social norm intervention studies aimed at reducing consumption of paper straws at one concession stand in a marine park. The first study was a shorter pilot, with the goal of narrowing down the type of social norm messages that would be included in the larger second study. The second study compared a control message, a positive injunctive norm message, and a negative injunctive norm message. Results indicated that the positive injunctive norm message significantly reduced paper straw consumption compared to the control condition. For each day of data collection, the authors collected a count of the number of drinks that were sold at the concession stand as well as a count of the number of paper straws that were requested. These values were transformed into a ratio and each condition was associated with a single value: the mean of these ratios across all days of data collection in that condition. Finally, the authors provided a cost-benefit analysis that extrapolated the results of this study to determine how much money this social norm intervention would yield the marine park over time.

I thought this manuscript was clearly presented and easy to read. The introduction has a nice overview of relevant literature and the discussion section thoughtfully brings up many other psychological factors that could be contributing to the results.

Response 1: Thank you for your comments. We have replied in detail below.

Major Comments:

Comment 1: The use of ratios obscures differences in base rates that may have varied by condition. For example, were there any differences in the number of drinks that were sold across conditions? Relatedly, was any other information about the day recorded? Number of visitors? Temperature? I’m wondering if it’s possible to rule out environmental factors that may have covaried with condition.

Response 1: We allocated the conditions randomly, so this should theoretically not impact our results. However, we recognize that our sample size for each condition is limited and so have added a sentence in the discussion to acknowledge this (Line 406): “It is important to acknowledge that we only tried one version of a message for each norm condition and that our sample sizes were limited, which means the randomization process may not have fully balanced all variables (e.g., respondent profile) across all conditions.”

Comment 2: I would be interested in seeing what the data that will be made available upon publication looks like. Will it simply be the averages of the ratios for each condition? Or will the data file include the number of drinks ordered and the number of straws requested for each day in each condition? I think the latter would be more useful for people interesting in further exploring this interesting dataset.

Response 2: The data we have available for each day includes the number of drinks ordered and the number of straws requested. You can now find the dataset here: https://figshare.com/s/648611d23cb654d0dfcf. The public DOI is: https://doi.org/10.6084/m9.figshare.15134754.v1

Minor Comments:

Comment 3: Page 2, line 30: “Concerningly, only 37.8% was recycled, while 45.7% was sent to landfills.” What accounts for the remaining 16.5%?

Response 3: We have updated these statistics with the latest published numbers from 2018 and have included percentages adding up to 100%. See Page 2, Line 49: “In 2018, European Union countries generated 2337 million tons of waste across all economic activities and households [2]. That is approximately five tons of waste per resident of the European Union. Concerningly, only 54.6% of waste was treated in recovery operations, including 37.9% that was recycled, while the remaining 45.4% was either sent to landfills, incinerated, or disposed of otherwise [2].”

Comment 4: Typo page 2 line 39: “Single-use plastics have recently become a become a hot button issue…” ‘become’ is repeated twice.

Response 4: We have corrected this typo. 

Comment 5: Page 8: “A message pair using positive and negative injunctive norms was selected for broader testing based on the results of the pilot, which showed that the mean ratio of straws taken to drinks sold was lowest for the negative injunctive social norm condition (x̄ = 0.214; SD = 0.041).” However, the results of the pilot study indicate that the negative injunctive social norm condition was not statistically different from any other condition (including the control condition). Might be more clear that the choice of the injunctive norms was based on the literature because the pilot study didn’t provide any clear winners.

Response 5: We agree. We have clarified the text starting on Line 246 to say: “Given the lack of a significant result in the pilot study, the choice to move forward with the injunctive message pair was mainly informed by studies that suggest that injunctive norms are more effective than descriptive norms at promoting desired environmental behaviors [28,29].”

Comment 6: Describe the nature of the operator error for both Study 1 and Study 2.

Response 6: We have added the following explanation to the text starting on Lines 202, “Operator error means that there were mistakes in the data collection on a particular day that rendered the data unusable. Errors included the implementation of incorrect message signs on a particular day, the temporary absence of straws in a dispenser due to shortages, and errors due to cashier shift turnover.” A brief explanation is also on Line 298.

Comment 7: Why do you use different statistical tests to answer the same question for Study 1 and Study 2? Study 1 uses a Kruskal-Wallis test to determine if there was a difference in ratios by condition. Study 2 uses a Welch’s F test to determine if there was a difference in ratios by condition. Either use the same test for each or provide a justification for why one is more appropriate for each case.

Response 7: We used a Kruskal-Wallis Test for the pilot because the data did not meet all of the assumptions for a parametric test, such as a one-way ANOVA. Two conditions had outliers and the results of a Shapiro-Wilk’s test demonstrated that the data were not normally distributed for two conditions. We have more clearly justified this decision from Lines 213 - 222: “The data did not meet the assumptions for a one-way ANOVA. Two conditions had outliers, as assessed by inspection of a boxplot: the negative injunctive norm condition had a single outlier, 

while the control condition had three outliers, two of which were extreme outliers (more than 3 box-lengths away from the edge of the box). The results of a Shapiro-Wilk’s test demonstrated that the data were not normally distributed for two conditions: control (p = .001) and negative injunctive norm (p = .034). There was homogeneity of variances, as assessed by Levene's test for equality of variances (p = .493). Because the data violated these assumptions, a Kruskal-Wallis test was run to determine if there were differences in ratios between the seven conditions. The mean rank of ratios was not statistically significantly different between groups, χ2(6) = 7.971, p = 0.240, ε2= .125 [41] This is a moderate effect size.”

Recognizing that a one-way ANOVA is somewhat robust to violations of the assumptions, we also ran a one-way ANOVA and found that there was still no difference between conditions. We have added these results on Line 222: “Recognizing that a one-way ANOVA is somewhat robust to violations of the assumptions [42] we also ran a one-way ANOVA and found that there was still no difference in ratios between the seven conditions, F(6,58) = 1.315, p = .265, est. ω2 = .029. This is a relatively small effect size.”

Comment 8: Page 11: When you inspect the normality of the distributions for each condition, are you looking at the distribution of ratio values over the 25-26 days for each condition? Regardless, make it clear what the values are that you are inspecting.

Response 8: Yes, we were inspecting the distribution of ratio values and have added this detail to the text to clarify. The sentence starting on Line 307 now reads: “The results of a Shapiro-Wilk’s test also demonstrated that the ratio values were normally distributed for the negative injunctive norm condition (p = 0.101) and the control condition, (p = 0.077), while the distribution of ratio values for the positive injunctive norm condition was on the edge of normal distribution (p = 0.05).”

Comment 9: Page 12 “In this case, we reject the null hypothesis and accept the alternative hypothesis that normative messages resulted in significantly less straw use than non-normative messages.” Should be more specific and say ‘a positive normative message’.

Response 9: We have made this edit in the text. The sentence beginning on Line 326 now reads, “In this case, we reject the null hypothesis and accept the alternative hypothesis that a positive normative message resulted in significantly less straw use than a non-normative message.”

Comment 10: Looks like the biggest intervention happened when the park decided that patrons had to ask for straws (a change that happened between Study 1 and Study 2). Would be good to discuss the effect sizes of the norm intervention in terms of the change in control condition values from Study 1 to Study 2.

Response 10: Thank you for this helpful comment. It is true that the effect sizes of the pilot and the main study are not directly comparable because of the change that happened between Study 1 and Study 2 where patrons had to start asking for straws. This change was beyond our control. We have more clearly acknowledged this limitation in the discussion section with a sentence starting on Line 469: “This change also means it is not possible to compare effect sizes between the pilot experiment and the full experiment, which is a limitation of the study.”

Comment 11: In Discussion section, be clear that you only tried ONE version of a message for each norm condition. It’s possible that different messages that fit within the different norm definitions may have impacted behavior differently.

Response 11: We have added the following sentences to the discussion, starting on Line 406 to address this point: “It is important to acknowledge that we only tried one version of a message for each norm condition and that our sample sizes were limited, which means the randomization process may not have fully balanced all variables (e.g., respondent profile) across all conditions. It is possible that different messages within the different conditions could have influenced behavior in other ways.”

Comment 12: The reactance suggestion in the discussion section would predict that negatively worded normative messages would encourage MORE straw consumption than the control condition. This did not happen.

Response 12: This is an interesting point and we agree that would expect reactance to run contrary to the effect of the norm. However, this could simply reduce the impact of the norm if the effect of reactance is smaller than the effect of the norm. The negatively worded normative message would only lead to more consumption if the effect of reactance was larger than the effect of the norm. We have explored this nuance further in the discussion and have also more clearly acknowledged that this explanation is speculative. See the paragraph starting on Line 419: “While we did not collect information on participant characteristics, which could help better explain our results, the psychological literature provides some potential explanations as to why the positive injunctive message may have been more effective. One possible explanation is that the negatively worded message may have triggered psychological reactance in some visitors [46]. Reactance is a form of motivational arousal that occurs when someone feels that their personal freedoms are threatened by rules and restrictions [46]. Resistance can occur when injunctive social norms (or proscriptive messages) appear to impinge on personal freedoms and may even lead people to increase the undesirable behavior [47]. While this is speculative, reactance may have decreased the effect of the negative injunctive message, effectively erasing any signal when compared to the control group.”

Comment 13: Page 16: “salience bias might also help explain why the negative injunctive message resulted in higher paper straw use” Be clear that you mean higher than the positive injunctive message, not higher than the control condition.

Response 13: Good point. We have made this clearer. This is the updated sentence starting on Line 447: “When comparing the positive and injunctive norm messages, salience bias might also help explain why the negative injunctive message resulted in higher paper straw use than the positive injunctive message [52].”

Reviewer #2: 

This paper described a simple experiment testing the effectiveness of normative messaging on the use of straws. Overall, it was clear and easy to read, however I felt the paper was missing many details to help us understand what was done, why, and the relevance of the findings. I outline these below.

Main concerns

Comment 1: Why is the data not provided to the reviewers? Data availability is required to publish in PLOS one and is highly relevant for the review process. Please make this available.

Response 1: You can now find the dataset here: https://figshare.com/s/648611d23cb654d0dfcf. The public DOI is: https://doi.org/10.6084/m9.figshare.15134754.v1

Comment 2: I would like to see the results of the pilot study below the study description. Understanding what was found in that experiment is important to understand the methods of the main experiment of the paper.

Response 2: We made this change as requested. 

Comment 3: The authors did not find any difference between conditions in the pilot study

Response 3: That is correct. We’ve highlighted this lack of difference starting on Line 220 of the results and again on Line 246: “Given the lack of a significant result in the pilot study, the choice to move forward with the injunctive message pair was mainly informed by studies that suggest that injunctive norms are more effective than descriptive norms at promoting desired environmental behaviors [28,29].”

Comment 4: Why use a Kruskal-Wallace test for the pilot? The non-parametric test was justified for the main experiment, but not the pilot. If it passes parametric assumptions, a two-way ANOVA would be appropriate (except for control condition), where you compare Positive/Negative manipulation, and manipulation type (descriptive, injunctive, social norm). Additionally, why not compare each individual group to the control, as you did in the main experiment? This result section was lacking, in both justifications and statistics.

Response 4: The goal of the pilot was to determine which treatment performed best overall, so we made a comparison between all conditions and the control, rather than comparing each individual treatment to the control. We have more clearly justified our use of the Kruskal-Wallis test by adding the following details, starting on Line 213: “The data did not meet the assumptions for a one-way ANOVA. Two conditions had outliers, as assessed by inspection of a boxplot: the negative injunctive norm condition had a single outlier, while the control condition had three outliers, two of which were extreme outliers (more than 3 box-lengths away from the edge of the box). The results of a Shapiro-Wilk’s test demonstrated that the data were not normally distributed for two conditions: control (p = .001) and negative injunctive norm (p = .034). There was homogeneity of variances, as assessed by Levene's test for equality of variances (p = .493). Because the data violated these assumptions, a Kruskal-Wallis test was run to determine if there were differences in ratios between the seven conditions. The mean rank of ratios was not statistically significantly different between groups, χ2(6) = 7.971, p = 0.240, ε2= .125 [41] This is a moderate effect size. Recognizing that a one-way ANOVA is somewhat robust to violations of the assumptions [42] we also ran a one-way ANOVA and found that there was still no difference in ratios between the seven conditions, F(6,58) = 1.315, p = .265, est. ω2 = .029. This is a relatively small effect size.”

Comment 5: All test decisions should be justified through the manuscript

Response 5: We have more clearly justified our test decisions throughout the manuscript. See Lines 213 - 222 for a justification of the Kruskall-Wallis test in the pilot study and Lines 304 - 319 for a justification of the Welch’s ANOVA in the main experiment.

Comment 6: What is operator error? Describe it in detail (can be in a supplement)

Response 6: We have added the following explanation to the text starting on Lines 202, “Operator error means that there were mistakes in the data collection on a particular day that rendered the data unusable. Errors included the implementation of incorrect message signs on a particular day, the temporary absence of straws in a dispenser due to shortages, and errors due to cashier shift turnover.” A brief explanation is also on Line 298.

Comment 7: Effect sizes should be interpreted throughout the manuscript

Response 7: Thanks for the reminder. We have added the following sentence to the results, starting on Line 317: “The mean ratios of paper straws taken to drinks sold were statistically significantly different between the different conditions, Welch’s F(2, 45.39) = 5.85, p = 0.006, est. ω2 = 0.112. This is a relatively large effect size.”

We have also added a sentence to the first paragraph in the discussion that interprets the effect size of est. ω2 = 0.112 for the main experiment. See the sentence starting on Line 392: “We tested whether normative messages could reduce visitors’ use of paper straws at a marine park and found that the positively worded injunctive message resulted in significantly less paper straw use than the non-normative message and that there was a relatively large effect size.”

Additionally, we have interpreted the effect sizes for the pilot study results, starting on Line 219. 

Comment 8: Data was collected over time, why was this not analysed? Was the straw to drink ratio consistent over time, or did it increase/decrease? This analysis is necessary to make any argument about effectiveness of the intervention. Without it, the discussion about savings over time is premature.

Response 8: We allocated the conditions randomly, and so would expect base conditions to be similar across treatments. However, we recognize that extrapolating the data from the experiment over a longer time frame is fraught with challenges, as our assumption that conditions will remain similar might not hold in practice. We have more clearly acknowledged these limitations starting on Line 501: “Our cost-benefit calculation is based on some key assumptions, including the assumption that all concession stands will sell similar numbers of drinks and the assumption that visitation rates or visitor behavior will remain similar. As the COVID pandemic has shown this is not always the case, however we feel that these estimates are still informative. Accepting these assumptions, our calculations project that the return on investment could be approximately 22% if the intervention ran for three years, and that rate would continue to increase over time.”

Comment 9: Why are there no comparisons to other stands in the park over the same time period?

Response 9: We were not able to collect the same data at other stands in the park over the same time period due to limitations of staff and funding.

Comment 10: The authors state this experiment is important because it is conducted in a European country and has been understudied, yet does not discuss cultural effects on messaging. This should be discussed in the introduction and in the discussion – it is very possible that the effectiveness of the messages (positive/negative) may be related to cultural factors

Response 10: This is an important point and we have added a sentence and reference to the introduction starting on Line 153: “It is important to study messaging in different countries and contexts because culture may influence how people respond to message frames [40].” We have also added a sentence to the discussion to address this point starting on Line 411: “Framing effects are also likely to vary across cultures and countries. When examining framing effects across seven Central and Western European countries, one study found differences in the way that consumers from different countries, even within the same region, responded to positive and negative message frames [40].”

Minor comments

Comment 11: Describe the Welch’s F test as non-parametric (line 243)

Response 11: The Welch’s F test is a parametric test. It is also known as Welch's ANOVA. It is an alternative to the one-way ANOVA and is used when the assumption of homogeneity of variances is violated. We have changed the name in the text to Welch’s ANOVA.

Comment 12: Interpret effect size of Welch’s F test (line 245)

Response 12: We have added an interpretation of the effect size after reporting the effect size starting on Line 317: “The mean ratios of paper straws taken to drinks sold were statistically significantly different between the different conditions, Welch’s F(2, 45.39) = 5.85, p = 0.006, est. ω2 = 0.112. This is a relatively large effect size.” 

Comment 13: Give actual numbers in Table 3, in addition to what is presented (number of drinks sold, number of straws taken)

Response 13: We have updated Table 3 to include this data. 

Comment 14: Acknowledge the discussion section on reactance as speculative

Response 14: Starting on Line 419, we changed the wording to more clearly indicate that the discussion section on reactance is speculative. “While we did not collect information on participant characteristics, which could help better explain our results, the psychological literature provides some potential explanations as to why the positive injunctive message may have been more effective. One possible explanation is that the negatively worded message may have triggered psychological reactance in some visitors [46]. Reactance is a form of motivational arousal that occurs when someone feels that their personal freedoms are threatened by rules and restrictions [46]. Resistance can occur when injunctive social norms (or proscriptive messages) appear to impinge on personal freedoms and may even lead people to increase the undesirable behavior [47]. While this is speculative, reactance may have decreased the effect of the negative injunctive message, effectively erasing any signal when compared to the control group.”

Editor Comments

Comment 1: I have now collected two reviews from two experts in the field, whom I thank for their detailed and thoughtful reviews. Both reviewers found the paper interesting and easy to read. However they both expressed a number of concerns that should be addressed in a major revision. Therefore, I would like to invite you to revise your paper following their comments. Needless to say that all comments should be addressed or rebutted. When uploading your revised manuscript, please upload also a response letter containing a point-by-point response to all the reviewers' comments.I am looking forward for the revision.

 Comment 2: Thank you for the opportunity to submit a revision. 

Journal Requirements:

Response: We have updated the formatting following the guidelines. 

Could you therefore please include the title page into the beginning of your manuscript file itself, listing all authors and affiliations

 Response: Yes, we have made this change. 

3. Please clarify in your Methods section that IRB waived the need for consent

Response: We have made this change, starting on Line 171: “The full experiment (Study 2) built on the results of Study 1 by testing the most effective social norm messages from Study 1 against a control message. This study was approved by the University of Florida’s Internal Review Board (IRB202002244) and the need for consent was waived.”

“This work was funded by national funds through Fundação para a Ciência e a Tecnologia (FCT) as part the project CIP - Refª UID/PSI/04345/2020 (Jean-Christophe Giger)”

Response: "The funders had no role in study design, data collection and analysis, decision to publish, or preparation of the manuscript." We have amended this statement in our cover letter and would appreciate it if you could change the online submission form on our behalf. 

Response: We have made the data available. You can now find the dataset here: https://figshare.com/s/648611d23cb654d0dfcf. The public DOI is: https://doi.org/10.6084/m9.figshare.15134754.v1

6. We note that Figure 1 in your submission contain map images which may be copyrighted. All PLOS content is published under the Creative Commons Attribution License (CC BY 4.0), which means that the manuscript, images, and Supporting Information files will be freely available online, and any third party is permitted to access, download, copy, distribute, and use these materials in any way, even commercially, with proper attribution. For these reasons, we cannot publish previously copyrighted maps or satellite images created using proprietary data, such as Google software (Google Maps, Street View, and Earth). For more information, see our copyright guidelines: http://journals.plos.org/plosone/s/licenses-and-copyright.

Response: We have obtained permission from the original copyright holder and have uploaded the form as an “Other” file with our submission. We have also changed the figure caption to read “Reprinted from Zoomarine under a CC BY license, with permission from Zoomarine.”

---

## [Decision Letter · Decision Letter 1]

9 Dec 2021

The effectiveness and efficiency of using normative messages to reduce waste: A real world experiment

PONE-D-21-25759R1

Dear Dr. Neves,

We’re pleased to inform you that your manuscript has been judged scientifically suitable for publication and will be formally accepted for publication once it meets all outstanding technical requirements.

Kind regards,

Xingwei Li, Ph.D.

Academic Editor

PLOS ONE

Additional Editor Comments (optional):

Reviewers' comments:

Reviewer's Responses to Questions

**Comments to the Author**

1. If the authors have adequately addressed your comments raised in a previous round of review and you feel that this manuscript is now acceptable for publication, you may indicate that here to bypass the “Comments to the Author” section, enter your conflict of interest statement in the “Confidential to Editor” section, and submit your "Accept" recommendation.

Reviewer #1: All comments have been addressed

---

## [Editor Report · Acceptance letter]

13 Dec 2021

PONE-D-21-25759R1 

The effectiveness and efficiency of using normative messages to reduce waste: A real world experiment 

Dear Dr. Neves:

I'm pleased to inform you that your manuscript has been deemed suitable for publication in PLOS ONE. Congratulations! Your manuscript is now with our production department. 

Kind regards, 

on behalf of

Prof. Dr. Xingwei Li 

Academic Editor

PLOS ONE